# Comparison of Cooled and Uncooled IR Sensors by Means of Signal-to-Noise Ratio for NDT Diagnostics of Aerospace Grade Composites

**DOI:** 10.3390/s20123381

**Published:** 2020-06-15

**Authors:** Shakeb Deane, Nicolas P. Avdelidis, Clemente Ibarra-Castanedo, Hai Zhang, Hamed Yazdani Nezhad, Alex A. Williamson, Tim Mackley, Xavier Maldague, Antonios Tsourdos, Parham Nooralishahi

**Affiliations:** 1School of Aerospace, Transport and Manufacturing, Cranfield University, Cranfield MK43 0AL, UK; nico.avdel@gmail.com (N.P.A.); t.c.mackley@cranfield.ac.uk (T.M.); a.tsourdos@cranfield.ac.uk (A.T.); 2Computer Vision and Systems Laboratory (CVSL), Department of Electrical and Computer Engineering, Laval University, Quebec City, QC G1V 0A6, Canada; ibarrac@gel.ulaval.ca (C.I.-C.); hai.zhang.1@ulaval.ca (H.Z.); Xavier.Maldague@gel.ulaval.ca (X.M.); parham.nooralishahi@gmail.com (P.N.); 3Department of Mechanical Engineering and Aeronautics, University of London, London WC1E 7HU, UK; hamed.yazdani@city.ac.uk; 4Mapair Thermography Ltd., Melbourn, South Cambridgeshire SG8, UK; Alex.A.Williamson@cranfield.ac.uk

**Keywords:** active infrared thermography, pulsed thermography, signal-to-noise ratio (SNR), UAV, aircraft-grade composites

## Abstract

This work aims to address the effectiveness and challenges of non-destructive testing (NDT) by active infrared thermography (IRT) for the inspection of aerospace-grade composite samples and seeks to compare uncooled and cooled thermal cameras using the signal-to-noise ratio (SNR) as a performance parameter. It focuses on locating impact damages and optimising the results using several signal processing techniques. The work successfully compares both types of cameras using seven different SNR definitions, to understand if a lower-resolution uncooled IR camera can achieve an acceptable NDT standard. Due to most uncooled cameras being small, lightweight, and cheap, they are more accessible to use on an unmanned aerial vehicle (UAV). The concept of using a UAV for NDT on a composite wing is explored, and the UAV is also tracked using a localisation system to observe the exact movement in millimetres and how it affects the thermal data. It was observed that an NDT UAV can access difficult areas and, therefore, can be suggested for significant reduction of time and cost.

## 1. Introduction

Active infrared thermography (IRT) is an infrared-based non-destructive testing (NDT) technique, which is used for a quick analysis of materials, structures, and components. The method is reliable due to its non-contact, qualitative, and quantitative inspection, regardless of the size and shape of the specimen of interest [1]. IRT evaluates materials without subsequently affecting the mechanical, physical, or chemical properties. However, the method is subject to interference when the environment is not controlled, or the experiment set-up is not optimal. Challenges such as non-uniform heating, low spatial resolution, and environmental noise cause some difficulties for defect detection and characterisation [2]. Inspections are now being used for many applications, but the experimental set-ups differ greatly. For example, the equipment is mounted on robot arms, and unmanned aerial vehicles (UAV), while the set-up conditions are not as favoured as laboratory conditions; therefore, it is likely the noise will increase due to the motion, vibrations, and sequence mismatching.

The afore-mentioned challenges call for the necessity of some signal processing methods in order to optimise the results from the inspection. Fourier transform (FT) is a commonly used image processing tool, which is used to decompose an image into its sine and cosine components [3]. This signal is filtered and the output image represents the image in the frequency domain, whereas the input image is the spatial-domain equivalent; each point will then represent a frequency contained in the spatial domain image [3]. In active thermographic signal processing, the method reads the thermal data sequence and extracts amplitude and phase images, which contain quantitative intelligible information [2].

LIT (lock-in thermography) and PPT (pulsed phase thermography) are two common NDT techniques used to evaluate the temperature response in the frequency domain using FT. The amplitude and phase angle data are computed from the temperature–time history of each pixel and are then stored as two-dimensional (2D) matrices. The matrices are converted into separate amplitude and phase images [2]. The amplitude image focuses on the surface of a sample up to a limited depth, whereas the phase image is less sensitive to artefacts such as non-uniform heating and emissivity variations, but focuses on the depth attained by the thermal wave; this can subsequently locate the defect depth [2].

The signal-to-noise ratio (SNR) in science and engineering can be defined as a measure that compares the level of a desired signal to the level of background noise [4]. In this context, SNR is conceptualised by comparing the signal located within a defected area of the material with the sound area of the material.

Mathematically, the SNR is the quotient of the (mean) signal intensity measured in the region of interest (ROI), i.e., the inhomogeneous area of the specimen, and the standard deviation of the signal intensity in an area outside the ROI [4]. Contrast-to-noise ratio is another commonly used scientific technique; it is similar to SNR and only differs by measuring the image quality based on a contrast rather than the raw signal [4]. The motive of this study relates to ongoing research of trying to integrate active thermographic NDT inspection with a UAV.

Serving the tremendous demand for high-performance, lightweight structures in various industrial sectors and applications, the use of carbon fibre-reinforced polymer (CFRP) composites and composite bonded joints is pivotal, and they play an immediate role for development of eco-friendly structures. Aircraft composites undergo continuous intense loads, and, after each cycle, they need to be free from any defect such as delamination, fatigue, and corrosion. Composites are prone to low-velocity impact damage, and this damage is not as critical for metallic structures. Impact damage can cause barely visible impact damage (BVID), which can quickly lead to a delamination that subsequently can reduce the component compression strength. This damage will magnify from the impact surface to the opposing surface and can potentially consequently damage reinforcing elements such as stringers [5]. Therefore, cost-efficient NDT is necessary during manufacturing and maintenance to ensure the safety of this complex material [6].

## 2. Thermal Technology

Thermal technology originated in military applications, because an IR sensor can produce a clear image in the dark, and it can also see through smoke, making this the ideal technology to locate opposing forces [7]. Figure 1 displays a brief timeline of the development of IR technology.

Thermal imaging technology was declassified in 1992 by the United States (US) government [7], which consequently allowed researchers and industry to further develop the technology and take advantage of its use in a variety of applications such as thermography, firefighting, and thermal weapons. Thermovision Model 661 is one of the first thermal cameras built; its total weight was approximately 25 kg, while that of the oscilloscope was 20 kg, and that of the tripod was 15 kg [8]. The operator also needed a 220-V alternating current (AC) generator set and a 10-L jar of liquid nitrogen to cool the camera [8]. In the 21st century, there has been a significant weight reduction and there are hundreds of cameras on the market weighing under 1 kg. Figure 2 shows a thermal IR camera known as the FLIR Boson.

This brief overview of thermal cameras really puts into perspective how far the technology has come. Thermal cameras are frequently used on unmanned aerial vehicles for a variety of applications, but there was little research using a UAV for active thermographic NDT. This research, comparing both cooled and uncooled cameras, stemmed from an NDT UAV project due to the fact that only an uncooled camera at the moment has the ideal size and weight requirements to mount on a UAV.

### 2.1. Infrared Imaging Systems

Optical systems, detectors, and the atmosphere are the three main subsystem elements of an infrared imaging system. The electronics and imaging processing subsystems support the detector system. A lens, a prism, and a mirror are the main components used within the optical system to collect and divert the optical path by reflecting and refracting the infrared radiation in order to focus all of the radiation on the detector focal plane array (FPA). The infrared detector is a sensor or transducer that is used to convert the signal that is proportional to the amount of infrared radiation incident on the detector FPA surface into an electrical signal, where the electronics will amplify the signal to a functional level. The digitalised signal will be subjected to image processing which can then be physically seen on a display. The atmospheric absorption of visible and near-IR radiation in the gaseous atmosphere is primarily due to $H_{2}O,\;O_{3},\;\mathrm{and}\;{\mathrm{CO}}_{2}$ [9]. The molecular vibration of these three molecules results in high absorption in various parts of the infrared spectrum. Therefore, detectors are optimized to pick up infrared radiation between specific wavelengths. The three atmospheric windows in the infrared spectrum are shortwave IR (SWIR; 1–3 μm), midwave IR (MWIR; 3–5 μm), and longwave IR (LWIR; 8–14 μm). Emitted radiation is transmitted through the atmosphere and optics of an infrared system, However, the fundamental question is how much radiation the infrared detector FPA absorbs [9].

### 2.2. Radiometry

The camera will receive radiation from the object of interest, combined with radiation from its surroundings that was reflected onto the object’s surface. Both radiation components become attenuated when they pass through the atmosphere. Since the atmosphere absorbs part of the radiation, it will also radiate some itself [10]. This combined radiation impinges the IR camera lens. Therefore, we can derive a formula to calculate the object’s temperature from a calibrated camera’s output, as shown in the illustrated image below (Figure 3).

The object emissions are equal to ε·τ $\lambda_{obj}$, where ε is the emissivity of the object and τ is the atmospheric transmittance. The reflected emission from ambient sources are equal to $\left(1-\varepsilon \right)\cdot \tau \cdot \lambda_{amb}$, where, (1−ε) is the reflectance of the object. The effective temperature of the object’s surroundings, also known as the reflected ambient temperature, can be written as $T_{amb}$, while $T_{atm}$ is the temperature of the atmosphere. It is assumed that the temperature $T_{amb}$ is the same for all emitting surfaces within the half-sphere seen from a point on the object’s surface. The atmospheric emissions are equal to $\left(1-\tau \right)\cdot \lambda_{atm}$, where (1−τ) is the atmospheric emissivity. The total radiation power received by the camera can now be written as $\lambda_{tot}$ = ε·τ·$\lambda_{obj}$ + $\left(1-\varepsilon \right)\cdot \tau \cdot \lambda_{amb}$ + $\left(1-\tau \right)\cdot \lambda_{atm}$.

In order to accurately gather the correct temperature of the object of interest, the IR software requires the object emissivity, the atmospheric temperature and humidity, and the temperature of the ambient surroundings to be input. These can be assumed, calculated manually, or found in available look-up tables, depending on the circumstances [12].

### 2.3. Cooled and Uncooled Cameras

Infrared cameras work in multiple ways; they detect infrared photons directly or they detect the small changes in temperature in an array of thermal elements. In infrared cameras, photovoltaic means that the infrared camera uses a material that produces a voltage difference when photons of a specific wavelength hit the material, whereas photoconductivity means that the infrared camera uses a material whose electrical resistance changes when a certain wavelength hits it. In thermal detectors, the absorbed radiation produces a temperature change in the detector itself; therefore, any physical property sensitive to temperature change can be used [11]. The specs of the two thermal cameras used in the active thermographic experiments in this paper can be found in Table 1.

All infrared cameras based on direct detection technology are detectors cooled to cryogenic temperatures close to 77 K. The photons captured are directly translated into electrons. The charge accumulated, the current flow, or the change in conductivity are proportional to the radiance of objects in the scenery viewed. The most common cooled system in scientific research is the InSb (indium antimonide) detector; this system collects the light within the 3–5-μm spectral band, consequently providing a better spatial resolution because the wavelength is much shorter than the 8–12-μm spectral band. InSb detectors have very good sensitivity, i.e., a low noise-equivalent temperature difference (NETD), as the elements tend to be smaller in size compared to the microbolometer detector elements; therefore, for the same required special resolution, InSb detectors require lenses with shorter focal lengths. A cooled camera has some drawbacks, as the systems are usually expensive with limited mean time between failures (MTBF), usually a few thousand hours. The systems are bulky and have a relatively long cooling down time [11].

Most uncooled cameras are based on the micro-bolometer. This sensor operates within an ambient temperature region and works by measuring minute changes in resistance, voltage, or current when heated by incident infrared radiation. The bolometer is a resistive element constructed from a material with a small thermal capacity and a positive temperature coefficient; therefore, the resistance changes with an increase in temperature. This change in resistance is as seen for the photoconductor; however, the detection mechanism differs. Radiant power produces heat within the material which subsequently produces a resistance change, but no direct photon–electron interaction takes place.

The introduction of uncooled cameras brought IR imaging to the mass market. Uncooled cameras rely on thermal detectors as opposed to quantum detectors, while recent advancements in semiconductor manufacturing and micromachining allowed for tiny intricate structures to be produced in large arrays at low cost, which drive today’s uncooled thermal detectors [13].

The varioCAM high resolution (HR) is designed for both stationary and mobile use for measuring and storing temperature data. Its low weight and long battery life give the camera extra versatility as opposed to the FLIR Phoenix which is heavy and connects directly to a power socket. On the other hand, the FLIR Phoenix has a slightly larger spatial resolution and a significantly higher NETD. Hence, it is expected to produce results with a higher SNR under similar inspection conditions.

## 3. Experimental Procedure

### 3.1. Composites

The specimens (samples 2.2. and 3.2) that were used in this study were of high-performance aerospace-grade uni-directional (UD) CFRP composite plies supplied by Hexcel, manufactured via pre-impregnation autoclave manufacturing according to the plies’ specifications (under 6 bar, vacuum bagged, and curing temperature of 180 °C). They were made of UD Toray 800S intermediate modulus carbon fibres and pre-impregnated with a high-performance rubber nanoparticle-toughened epoxy matrix, HexPly^®^ M21, which was designed to exhibit excellent damage tolerance. HexPly^®^ M21 was developed as a controlled flow system to operate in environments up at 121 °C (250 °F) [16]. Such composites have an excellent strength-to-weight ratio, which led to a rapid rise in the usage of the material in the aerospace industry. The plies were stacked in a quasi-isotropic, symmetric configuration to represent a fuselage skin laminate [17].

Impact damage on an aircraft can occur at any point during operation; however, it is more likely to occur during take-off or on the ground during maintenance and handling operations. Furthermore, 87% of total composite damage is caused by impact, with energy ranging from 10 to 100 J from mechanical collision, ranging from a dropped tool or maintenance vehicle impact. These low-impact damages usually lead to barely visible impact damage (BVID) [5]. Table 2 displays the common examples of impact damage cases on commercial aircraft.

In the past two decades, there were about 90,000 bird strike impacts in the United States of America (USA) alone, this is an example of how common aircraft can be subject to damage. A bird strike is representative of a low-velocity rigid impact scenario, which represents the damage under inspection in the composites seen in Figure 4. The repercussions of aircraft damage can lead to a significant financial loss, due to associated costs such as repairs, as well as delays and cancelation of flights [5].

The curing followed the preconisation from Hexcel^®^ datasheet [16], at 180 °C and 7 bar for 120 min, with an initial heating ramp of 1 °C per minute. Samples were then sent to be cut by water jet to obtain very precise dimensions. The sample’s dimensions were 150 mm × 100 mm. Different characteristics of the cured M21composite are shown in Table 3, obtained from Reference [18] with G_IC_ and G_IIC_.

### 3.2. Impact Testing

To simulate an impact damage similar to what an aircraft would be prone to, the CFRP samples were impacted using the Imatek IM10 drop tower from the Cranfield Impact Centre (CIC). The machine is equipped with a rebound catcher device to prevent multiple bounces. It consists of a falling weight tower, several meters high, with a frictionless guide rail where the impactor can slide. The impactor is made of a core with rolling bearing, a Kistler 20-kN piezo-electric strain gauge load cell, and a hemispherical striker of 16 mm, for a total mass of 4.06 kg.

The drop tower which can be seen in Figure 5 has a guide rail with laser triggering for the velocity sensor and a rebound catcher mechanism in order to accurately simulate impact damage. The force-versus-time data during contact were recorded from the strain gauge every 0.05 ms, corresponding to a frequency of 20 kHz, and the software calculated the striker’s acceleration, displacement, and velocity during the test, as well as the energy transferred to the sample and given back to the striker during the bounce. The specimen to be tested was maintained by four clamps on a plate with a rectangular hole of 125 mm × 75 mm.

## 4. Results and Discussion

The CFRP sample 2.2 was subjected to 24 J of impact, whilst sample 3.2 was subjected to 8 J of impact. Both samples were inspected using active thermography with both a cooled and an uncooled system simultaneously. The data were then analysed using signal processing in order to further assess the damage and then compare the datasets from each IR camera system by means of SNR.

The method strategy can be seen in the flow chart below. The methods used are commonly used methods within the scientific community; therefore, they are only briefly described throughout this paper. The method strategy used in this research can be seen in Figure 6 as a flow chart.

### 4.1. Signal Processing

The datasets from the cooled and uncooled systems are subject to multiple signal processing techniques. The emissivity of the composite samples can influence the accurate mapping of thermal patterns; it can lead to illusory temperature inhomogeneity of infrared images, which results in influences on detecting defects. There are various ways to reduce surface emissivity influence such as paint or tape; however, this is not always feasible due to contamination of the specimen of interest. Even though, samples 2.2 and 3.2 were black and made of carbon composites, they had a relatively high reflectance. Therefore, the thermal image could show corrupted data as some of the electromagnetic radiation from the high-intensity flash and the environment (e.g., people moving in the surrounding area) would be reflected and the energy emitted from the sample is not optimal. The conditions were still extremely good (under laboratory conditions without any significant interference) to perform an inspection, and parasite reflections could be corrected during post-processing [19].

A method called “cold image subtraction” using ir_view software (Visooimage Inc.) is a signal processing technique which removes fixed patterns such as noise and fixed environmental reflections. Figure 7A of the sequence is in a steady state (before excitation), while, in active IRT, the cold image is a thermogram acquired prior to the heat pulse. All pixels from the sequence with the same intensity as the cold image are removed; therefore, frames without stimulation contain just noise. The pixel values are increased due to the excitation; at this point in the sequence, the images have less noise (e.g., reflectivity) due to the subtraction.

#### 4.1.1. Defect Analysis

Thermographic signal reconstruction (TSR) is a useful processing technique in pulsed infrared thermography as it uses surface temperature evolution based on a one-dimensional solution of the Fourier equation for a Dirac delta function in a semi-infinite isotropic solid as shown in Equation (1) [20,21]. The temperature decays can be fitted with a functional relationship. Shepard et al. [21] proposed such fitting several years ago; it is known as the TSR (thermographic signal reconstruction) algorithm, with which thermal data from pulsed thermography experiments are fitted using a logarithmic polynomial function [22]. This method was described well by Duan [20] for automated defect classification in infrared thermography based on a neural network [20].
(1)ΔT=T(0,t)−T0=Qeπt,
where *T*0 is the initial temperature, *e* is the material thermal effusivity, *Q* is the energy density absorbed by the surface, and *t* is the time after excitation.

Equation (1) in the logarithmic domain can be expressed as shown in Equation (2) [21].
(2)ln(ΔT)=ln(Qe)−12ln(πt).

This corresponds to a straight line with a slope of −0.5 on a log–log scale as seen in Figure 8.

The essential thermal response is preserved by applying a low-order expansion using Equation (2) to serve as a low-pass filter. In the logarithmic domain, including higher orders replicates noise [20].

After TSR, defective regions usually appear with improved contrast.

An *n*-degree polynomial is fitted for each pixel within the region of interest, Typically, *n* is set to 4 or 5 to avoid “ringing” and ensure good correspondence between data and fitted values. Using MATLAB, a direct polynomial fitting can be applied. The advantages of this synthetic data processing are the significant reductions in noise, while less storage is required, and it allows for analytical computations such as PPT. A five-degree polynomial is an excellent fit for pulsed thermography data, which can be given as follows [20]:(3)ln(ΔT)=a0+a1ln(t)+a2[ln(t)]2+a3[ln(t)]3+a4[ln(t)]4+a5[ln(t)]5.

The presented graph in Figure 9 is constructed from the defected and non-defected area of a pixel. It follows the typical monotonic decay due to surface cooling of the samples. TSR processing commonly amplifies cooling curve differences through a logarithmic operation. TSR provides significant data compression because it only requires saving six polynomial coefficients per pixel [20,21]. Figure 9 is an example of a one-dimensional approximation; this recognizes that heat diffuses mainly in one direction. Therefore, it assumes that lateral diffusion components, more or less, cancel out in a defect-free setting. In the presence of a non-homogeneous subsurface boundary, incident heat flow from the sample surface is impeded [20,21].

The fourth frame in dataset 2.2CooledFront is where the electromagnetic radiation from the flash hit the specimen, and the damage can be seen instantly. Figure 9D is a temperature/time graph, where the red signal represents a pixel within the defected area, and the blue pixel represents a pixel in a sound area (undamaged). Each black dot along the plot represents a frame within the sequence. The peak of the signal is where the high-intensity flash hit the surface of the specimen. A logarithmic scale is used to improve visualisation. There is an obvious signal difference as the red (defected) signal is significantly more temperature intense than the blue (sound) signal.

#### 4.1.2. Pulsed Phase Thermography (PPT) (Fourier Transform)

PPT transforms the data from the time domain into the frequency domain using one-dimensional discrete Fourier transform. The phase of the Fourier transform was computed using MATLAB. After processing, the lowest frequencies (first frames) usually display the deepest defected areas. Figure 10 displays the 2.2CooledFront sample data processed using PPT.

#### 4.1.3. Principal Component Technique (PCT)

PCT is a technique used to identify patterns in data, such as the similarities and differences, whilst also compressing the data without losing much information. Using MATLAB, the sequence of images was computed by PCT. Unlike FFT, this technique does not identify the depth of the defect, but it is a powerful tool to qualitatively analyse the data statistically [23]. Figure 11A displays the *2.2CooledFront* sample data processed using PCT, whereas Figure 11B was processed using PCT but the contrast levels are also adjusted for visual aid.

#### 4.1.4. Signal-to-Noise Ratio (SNR)

In scientific literature, there are multiple ways to calculate the SNR. Below is an example of seven widely used SNR equations in the scientific community [24].
(4)$${\mathrm{SNR}}_{1}=\frac{\mu s}{\sigma N};\;{\mathrm{SNR}}_{2}=\frac{\left|\mu s-\mu N\right|}{\sigma n};\;{\mathrm{SNR}}_{3}=10{Log}_{10}\frac{{\left|\mu s-\mu N\right|}^{2}}{\sigma_{N}^{2}};\;{\mathrm{SNR}}_{4}=\frac{\sigma s}{\sigma N};\;{\mathrm{SNR}}_{5}=\frac{\sigma_{s}^{2}}{\sigma_{N}^{2}};\;{\mathrm{SNR}}_{6}=10{Log}_{10}\left(\frac{\sigma_{s}^{2}}{\sigma_{N}^{2}}\right);\;{\mathrm{SNR}}_{7}=\frac{\left|\mu s-\mu N\right|}{\sqrt{\left(\sigma_{s}^{2}+\sigma_{N}^{2}\right)}/2}.$$

In these equations, *μs* is the average level of the signal in the defect region of the image, *μN* is the average level of the noise in the reference or sound region of the image, *σs* is the standard deviation of the signal in the defect region of the image, and *σN* is the standard deviation of the noise in the reference or sound region of the image.

The arithmetic mean of the pixel intensities in the corresponding regions is equal to the average level. The standard deviation (STD) is described as the noise, which includes all noise ignoring the specific influence of each individual noise [24]. The STD is defined as the square root of the mean of variances [24]. According to the European standard for radioscopic testing, the noise is described using the root mean square (RMS), which is generally defined as the square root of the mean square [24]. If the average level is zero, the STD and the RMS are equal; therefore, they are interchangeable when it can be assumed the input signal has zero mean [24]. The noise is defined as RMS of pixel values in the thermal data, in both the sound and the defected area.

${\mathrm{SNR}}_{1}$ translates to the ratio of the signal level to the noise [24]. ${\mathrm{SNR}}_{2}$ describes the signal power by introducing the contrast. ${\mathrm{SNR}}_{2}$ has an analytical relationship with ${\mathrm{SNR}}_{1}$ which can be expressed as ${\mathrm{SNR}}_{2}$ = ${\mathrm{SNR}}_{1}$ if *μN* = 0 ∧ *μs* > 0. In other words, when the average level of the noise is zero, the two are equal, which would occur when the background is subtracted from the image. ${\mathrm{SNR}}_{2}$ can be described as contrast-to-noise ratio, which is frequently used in scientific literature. The contrast is calculated using the absolute value because, in general, a positive or a negative contrast between the defect and the image background is considered equal. As a result of signal processing techniques such as TSR, or PCT, negative contrast occurs, which is where the background level is above the defect [24].

${\mathrm{SNR}}_{3}$ is similar to ${\mathrm{SNR}}_{2}$ but converted to decibels, which is common in signal processing. The squared arguments in the equation follow the conversion from voltage to power in electronics, where the energy or power is proportional to the square of the amplitude. When the SNR is expressed in decibels, negative values can appear as log(*x*) < 0 ∀*x* ∈ (0, 1). The analytical relationship between ${\mathrm{SNR}}_{3}$ and ${\mathrm{SNR}}_{2}$ can be expressed as ${\mathrm{SNR}}_{3}$ = 20 log_10_ (${\mathrm{SNR}}_{2}$) [24]. ${\mathrm{SNR}}_{4}$ is defined as the ratio of standard deviations, and ${\mathrm{SNR}}_{5}$ is described as the ratio of variances. The dispersion from the mean in both ${\mathrm{SNR}}_{4}$ and ${\mathrm{SNR}}_{5}$ is what the power of the signal’s calculation is based on. ${\mathrm{SNR}}_{6}$ can be expressed as ${\mathrm{SNR}}_{5}$ = (${\mathrm{SNR}}_{4}$)2, ${\mathrm{SNR}}_{6}$ = 10 log_10_ (${\mathrm{SNR}}_{5}$).

In all these definitions, the power of the signal and the noise is calculated using the standard deviation [24]. ${\mathrm{SNR}}_{7}$ measures the power of the noise whilst considering the standard deviation of the signal. The average between the standard deviation of the signal and the standard deviation of the noise is estimated as the noise in ${\mathrm{SNR}}_{7}$ [24].

The raw and TSR data from samples 2.2 and 3.2 from both the cooled and the uncooled IR cameras were subjected to the seven SNR calculations. The results aim to display the difference in unwanted interference acquired by each camera. Figure 12A demonstrates the SNR techniques being calculated. The red square represents the damaged area, whereas the blue square represents a random undamaged area. Figure 13 displays the results from each of the seven algorithms.

The SNR magnitudes change greatly from one definition to the other, and the values are presented on the y-axis of the graphs in Figure 13. Nevertheless, the behaviour of each SNR through time can be compared graphically. For instance, all SNRs show a maximum around frame 24 with the exception of SNR7, which shows it much later (for this particular defect).

### 4.2. Active Infrared Thermography

The phase and PCT data were subjected to TSR, which ultimately resulted in smoother data. The frequency of both cameras for the inspection was set at 40 Hz. Following the data processing techniques, the contrast was also adjusted for a cleaner image. The images were magnified to display the true extent of the damage.

In Figure 14A, the damages are enlarged for simplicity; therefore, in Figure 14B, the damage size is displayed in centimetres to show accurate dimensions. The camera needs to be calibrated prior to inspection and mounted exactly parallel with no skew. A known size in the image can then help calibrate the pixels per metric. In this case, the size of the sample was known; therefore, we could use that reference for calibration and then find the size of the damage by outlining it within MATLAB. The *x*-axis displays the image frame from the sequence, and the *y*-axis is the pixel intensity. Figure 15 provides the results and comparison from all datasets.

### 4.3. Signal-to-Noise Ratio Comparison

The data for the uncooled and cooled inspections were analysed simultaneously so that the conditions remained identical; however, due to this, both cameras were at slightly different angles and distances from the specimen. VarioCAM had conditions that were more unfavourable, i.e., lower spatial resolution (number of pixels), lower thermal resolution (NETD), and larger field-of-view (FOV); therefore, there were a smaller number of pixels in the region of interest. However, using the results, we can determine to what extent a lower-end camera (VarioCAM) could perform adequately or not, when compared to a higher-end camera (Phoenix). Figure 15 displays the calculated seven SNR equations from each dataset. The mean areas of the defect and sound areas are available but not included for simplicity.

The numerical values were output after the SNR calculations from the pixel intensity regions selected within the defect and sound areas. The shaded boxes display the higher SNR value comparing both the cooled and uncooled dataset. To compare these data objectively using the numerical values, the cooled sensor displayed the best results, which was expected as it is a more sensitive IR system. On sample 2.2 Back, the uncooled system was clearer in terms of contrast than the cooled system but just not as sharp with regard to image quality. All in all, the uncooled sensor performed well for the larger defect, but not as well for the smaller defect.

On sample 3.2UncooledBack, ${\mathrm{SNR}}_{4,5,6}$ picked up a maximum SNR at frame 199, which has a greater value than the cooled system; however, this was actually due to reflection which is visible in that frame, resulting in a contrast difference. This demonstrates how radiation interference can affect datasets. Figure 16C displays the noisy pixel map of Frame 199.

Figure 16A displays the cooled three-dimensional (3D) pixel data of frame 40 (results from ${\mathrm{SNR}}_{2,3}$), which clearly shows a much smoother surface as opposed to frame 11 from the uncooled sensor. TSR is necessary to eliminate such noise.

Overall, the cooled system was consistent throughout when locating the maximum SNR, as ${\mathrm{SNR}}_{2–6}$ were all consistently the same frame. The frames varied slightly more in the uncooled system. The purpose of this paper is not to compare the individual SNR algorithms but to display the effectiveness of a cheaper, smaller uncooled IR system since such hardware is needed for active thermographic NDT with a UAV. The several different algorithms are helpful to compare and analyse the behaviour and ensure there was no algorithm bias.

#### Interpretation

The SNR equations were used to measure the SNR of the afore-mentioned NDT data. The data were raw with 500 total frames, including frames before the pulse and the cooling down phase. The defects produce a thermal contrast, where the SNR measures each defect in every frame using the selected regions of the damaged and sound areas. The algorithm searched for the maximum SNR in each frame. Figure 15 displays the frame including the corresponding calculated value. The output graph displaying the behaviour was previously explained and shown in Figure 13. All the graphs are available but not displayed for simplicity reasons.

${\mathrm{SNR}}_{1}$ depends on the average background level, and the results displayed were significantly different than the others, thus incorrectly quantifying the defect.

${\mathrm{SNR}}_{2}$ through to ${\mathrm{SNR}}_{6}$ all agreed on identifying the time at which the maximum SNR occurred on all the data from the cooled camera.

In the uncooled SNR dataset, ${\mathrm{SNR}}_{2}$ and ${\mathrm{SNR}}_{3}$ agreed with each other and ${\mathrm{SNR}}_{4}$, ${\mathrm{SNR}}_{5}$, and ${\mathrm{SNR}}_{6}$ agreed with one another. From this, we can understand that the cooled system provided reliable data as five of the SNRs were consistent, whereas the uncooled system seemed to be more unpredictable. That being said, in Sample 2.2 (24 J) ${\mathrm{SNR}}_{2}$ and ${\mathrm{SNR}}_{3}$, which are commonly used for NDT thermography, a higher SNR was shown than that in the cooled system; however, as previously mentioned, it still lacked in image quality. Usually, the time at which the maximum SNR appears increases with the depth of the defect. Moreover, a deeper defect results in a lower maximum SNR [24].

The SNR dataset 3.2 Front showed a significant difference between the cooled and uncooled results. All the SNRs for the uncooled data failed to discover the true extent of the present defect on the sample; however, it remained adequate as it still located the damage somewhat (see Figure 17).

In 3.2Back, ${\mathrm{SNR}}_{2\;\&\;3}$ successfully found the defects. The cooled data showed the defect, but it was difficult to see, whereas the Uncooled system showed the defect with brighter contrast. ${\mathrm{SNR}}_{4,5,6}$ successfully found the defect in the cooled dataset, whereas, in the uncooled dataset, they instead located the frames where the reflection was present. ${\mathrm{SNR}}_{7}$ successfully located the defects, with sharper images than the others.

To summarise, excluding ${\mathrm{SNR}}_{1}$, all the other SNR methods successfully located the defects when tested on the cooled data, while Sample3.2Back was not as clear as the others but still visible. The uncooled sensor struggled to adequately locate the smaller defect (8 J). The front surface of sample3.2 was further examined. The region of interest was defined to display a focused image and reveal the defect (see Figure 17A). This defect is the outline of the drop tower impact, which is visible with the human eye. The maximum SNR was due to this contrast difference for this image, but the noisy dataset was also a contributing factor, which can lead to missing the true extent of the damage. Figure 17B shows the 3D pixel graph, which displays the noisy and inhomogeneous dataset, making it difficult to locate the defect.

The SNR can be improved when calculating the data after it is subjected to TSR; then, the defects show improved contrast and sharpness, which benefits the data containing defects with greater depths. Figure 18 shows the SNR after TSR, which now follows the same behaviour as the other uncooled datasets, where ${\mathrm{SNR}}_{2,3}$ and ${\mathrm{SNR}}_{4,5,6}$ are consistent. The images are less noisy, and the defect is clearly visible.

SNR is commonly used for validation among the scientific community, especially in NDT IR thermography work. Zhang [22] used SNR to validate results from a pre-processing proposed modality IRT method. The pre-processed method is different from the usual signal smoothing modality which only uses a polynomial fitting as the pre-processing method, whereas the proposed modality used the low-order derivatives to pre-process the raw thermal data alongside using advanced post-processing techniques such as PCT and PPT. The effectiveness of this modality was verified for PT. It was found that the method provided better performance than lock-in heating in PT, in which the presented pre-processing modality cannot be used because the thermal curves for lock-in heating are not linear [22].

Shrestha and Kim [25] demonstrated how SNR differs when the frequency of the excitation source changes. The research compared different frequencies to discover and achieve the best SNR with regard to excitation frequency [25]. The work combined amplitude and phase images to enhance the SNR of inclusions. In the research, PCT and DWT (discrete wavelet transform) pixel level data fusion methods were performed, and SNR was introduced to verify the robustness of the results. The resulting fused images showed that PCA-based data fusion significantly enhanced the SNR of inclusions, especially in the higher excitation frequency range where inclusions were detectable. The differences in SNR between the input amplitude and phase signal were low, as compared to a low excitation frequency, where the differences in SNR between the input amplitude and phase signal were relatively high. However, it was also noted that the DWT algorithm worked well for the low excitation frequencies as compared to higher ones [25].

The SNR is affected by parameters such as the depth, diameter, material properties, or type of the defect. The influence of the parameters can be understood on a case-by-case basis; as experimental set-ups differ, the SNR is affected. The key is to understand the properties of the material thermal conductivity and density. For example, when the thermal conductivity is high, the resulting thermal contrast is also high. Thus, the SNR is very good [25].

Usamentiaga et al. [25] demonstrated the reliability of PCT for providing good results; however, while skewness and kurtosis provided good results, they failed to identify some defects. Therefore, a lot of small defects would go undetected without data processing techniques, which is proven to increase the SNR. The results showed a correlation between the depth of the defect and the SNR; in general, a deeper defect led to a lower SNR. This is a known issue of non-destructive thermographic inspection because the detection of deeper defects requires more energy [25].

Comparing data processing techniques is difficult, because, at the moment, there is no single method that maximizes the SNR for all materials, nor is there a single data processing technique that maximizes SNR and the defect detection rate. The selection of a data processing technique depends on the particular application and experimental set-up [25].

## 5. UAV Inspection

An NDT test was performed using the DJI M210 (Figure 19A) equipped with a red/green/blue (RGB) and thermal sensor (Figure 19B), weighing a total approximate weight of 3.84 kg with two batteries. The thermal camera was radiometric, with a 13-mm lens and 640 × 512 pixels, which is identical to the afore-mentioned FLIR Phoenix and slightly higher resolution than the VarioCAM.

### 5.1. Experiment

The UAV and the composite sample were tracked using a Vicon localisation system. This tracks the objects in 3D space using a series of cameras, displaying the exact UAV movements whilst operating indoors without the use of a global positioning system (GPS). Figure 20 shows the UAV flying indoors with images of the VICON system software tracking the UAV.

### 5.2. Localisation

The location data were tracked in order to analyse the stability of the UAV whilst capturing the data sequence. The calculated distances display how much the UAV moved during the data capture. An excess of movement can seriously affect the quality of the data.

Whilst capturing the thermal sequence, the UAV moved a total of 198 mm along the *x*-axis, which resulted in images being taken at different distances from the sample, consequently resulting in a noisy sequence and frames with alignment issues.

From the *x*-axis, we can also calculate the distance of the UAV from the wing when the data were captured using the formula below.
(5)sL (Specimen Location)−uL (UAV Location)=D (Distance between UAV and Specimen)(−2971mm)−(−1678mm)=1293mm.

To get the correct movement of the UAV, the flight envelope is calculated in 3D space.
(6)Flight Envelope:  dS−dE=Fe,
where *dS* is the position at which the data capture started, *dE* is the position at which the data capture finished, and *Fe* is the flight envelope movement during data capture.

Figure 21, Figure 22 and Figure 23 show the data displaying the full movement of the UAV in 3D space along with the methods to translate the localisation data received from the Vicon system into interpretable measurements. For optimal NDT results, the camera needs to remain as static as possible in order to reduce noise.

An autonomous flight brings more stability to the inspection. However, for any movement that cannot be avoided, the Vicon system helps provide the exact movement data in the specific environment. The autonomous flight helps keep the *x, y, z*-position in 3D space; however, the roll, pitch, and yaw of the UAV itself may not be as consistent. The UAV’s flight controller usually keeps making small adjustments so the UAV can maintain its 3D position. The consequence of this would result in slight misalignment in the image data in terms of skew. However, if the UAV is equipped with a gimbal, this will counteract any minor sudden movements; therefore, the images will be aligned, and the data will be valid. The localisation is more of an indoor problem, as access to GPS allows the UAV to maintain its position outdoors. There are also RTK (real-time kinematic) systems for many off-the-shelf UAVs which are available; these increase accuracy and are extremely reliable. With outdoor inspections, there are other challenges as briefly described in Section 5.4.

For indoor localisation, an onboard embedded system can connect to the UAV via serial and attitude commands, which can be sent to the aircraft allowing for accurate indoor positioning. Additionally, other onboard sensors can assist, such as stereo vision, ultrasound, and light detection and ranging (LiDAR). There are a few different ways to accurately maintain a UAV position indoors. SLAM (simultaneous localisation and mapping) is a common method; however, it is computationally heavy. Using a virtual remote control (RC) is another, where the UAV can be controlled through the serial port by simulated channel values; for example, the throttle can be adjusted by increasing the specific values manually. The DJI OSDK (onboard software development kit) allows for this type of control. The safest most accurate approach would be to use a localisation system such as Vicon. Vicon can be used as a relative 3D environment, where the drone can be tracked as described previously, and these values can be fed back into the flight controller using the DJI OSDK. The GPS is now being replaced with this new positioning system. Figure 24 displays more information regarding the UAV, onboard computer (OC), and the Vicon system set-up.

### 5.3. NDT Indoor Concept

The composite wing was manufactured at the Cranfield composite lab, with the knowledge that it contains no defects. However, to demonstrate the UAV NDT inspection, debris was placed within the wing to demonstrate an abnormality. Using a low-temperature heat gun, the material properties were stimulated. The UAV was flown manually without any GPS and data of the wing were captured.

Due to the UAV not maintaining its 3D exact position, the temperature/time graph in Figure 25E displays a wide range of pixel values across a short period of time. This occurred due to the thermal camera not remaining in a static position whilst capturing the data, resulting in unaligned frames. An initial abrupt rise and subsequent cooling curve should be expected, as seen in the afore-mentioned active thermographic test in Figure 9D. Figure 26 shows the pixel line profile over the “defected” area.

### 5.4. NDT Outdoor Concept

Gathering adequate data of composite aircraft in real environments is a challenge due to the legality and the inaccessibility of the relatively new expensive aircraft. However, a retired Boeing 737-400 was inspected using a UAV equipped with a thermal and RGB camera. Most of the valuable data were captured using the RGB camera; however, the IR camera gave us access to data that were not visible with the human eye or the RGB camera.

Note that this aircraft is predominantly made from aluminium; however, there are some composites within the aircraft, such as the flight surfaces. The following data are not a good representation of the portrayed NDT method described in this research, as, when a composite is damaged, it reacts differently to aluminium. Aluminium also possesses a much lower emissivity.

Figure 27A is an IR image of the aluminium fuselage, where the stringers and struts (frame of the aircraft) are visible, and the four dots are aluminium stickers for reference points only (see Figure 27B for corresponding RGB image). One of the main challenges of active thermography is getting a suitable excitation source. The B737 was situated in an airport where there was no access to a power supply. The excitation source used to gather these data was a portable Quarts IR heater. This sequence was subjected to the afore-mentioned signal processing methods. Using these methods, some abnormalities became present. Barely visible rivets that could not be seen in the initial IR data or in the RGB image due to them being hidden under the paint could in fact be picked up and displayed after processing. These rivets where inhomogeneous with respect to the skin of the aircraft and the other rivets, which is why these became visible and not the other rivets. Although this is not a representation of a damaged fuselage, it gives insight into the effectiveness of the post-processing techniques, especially when looking for small defects such as lightning strikes. Only specific parts of this aircraft could be inspected using active thermography due to the immobility of the excitation source. For inspection of aerospace aluminium structures, Ciampa et al. [27] compared multiple signal processing methods where damages were successfully located.

A mobile versatile excitation source needs to be developed or integrated onto a UAV to allow such an inspection to be used to its maximum potential. Adequate excitation sources are usually heavy and consume a lot of power, which makes it difficult to integrate them onto a UAV. Although these results did not demonstrate the location of subsurface composite damage, they established the use of a UAV equipped with a thermal camera outside (uncontrolled environment) where there are other external parameters and increased interference.

When using a UAV equipped with an IR camera for detecting damage outdoors, there will likely be a temperature difference. Overall, this might affect the data under extreme environmental conditions. Extreme conditions (high winds, etc.) can impact the data. However, under relevant normal conditions, active thermography is unlikely to be affected, whereas passive thermography would be, as this does not perform extra stimulation of the material. When using active heating, the assumption is that the defect signature will be significantly larger than environmental artefacts such as reflections, etc.

The UAV used a GPS to keep its position outdoors, which eliminated the localisation issue in Section 5.2, allowing Figure 27B to be successfully post-processed using methods such as TSR.

Finally, since there are some composite parts on the B737, the UAV was used to inspect the tail of the aircraft using the IR and RGB cameras. The top of the tail was just above 11 m from ground level. Therefore, it was not possible to use an external excitation source at this moment in time; however, in this instance, the sun naturally excited the surface during the day. The IR data in Figure 28A display some subsurface abnormality beneath the aluminium panel on the tail, while Figure 28B displays the corresponding RGB image. The abnormality will require further industry-renowned NDT inspection such as ultrasound methods to reveal the true extent of the damage; however, this is a difficult area to access without the use of a cherry picker or UAV. The rudder is made from composites (graphite/epoxy), and the internal structure is visible from the IR data; however, no concerning damages were located using just passive thermography.

## 6. Conclusions

The study contributes to reducing the time and cost of NDT inspections for composite aircraft. The research displays how there were significant improvements in thermal technology and provides the theory of how cameras operate on a scientific level. The study demonstrates how difficult it is to get optimal results due to the noise and interference that thermal cameras are prone to, while it also shines a light on the challenges faced when using a UAV for NDT purposes.

The focus was demonstrating the effectiveness of two thermal cameras, the FLIR Phoenix, which is a high-end cooled system, and the VarioCAM, which is a lower-end uncooled system. Due to uncooled cameras being small, lightweight, and cheap, they are more accessible to use for UAV applications. Therefore, this work compared the uncooled and cooled thermal cameras by exploring the signal-to-noise ratio data from an active thermographic NDT of aerospace-grade composite samples. The uncooled system performed adequately when locating the defects (impact damage) and when subjected to the SNR calculations. In one dataset, where the SNR failed to locate the defect adequately, the use of thermal signal reconstruction before the SNR calculations helped fix the maximum SNR problem as it then successfully located the defect. This proved that the lower-end camera is not up to the same standard in terms of resolution and all-round performance; however, with some further processing, it was possible to attain acceptable results, meaning that the uncooled camera would suffice for such IR thermographic testing.

The SNR compared the two cameras using well-known scientific equations. However, the active IR thermographic results from post-processing displayed considerably improved results from both cameras, successfully locating the full extent of the damage subsurface and on the surface, which cannot be seen with just the human eye. The SNR, no matter what definition is used, is independent of the material type. This concept was used to evaluate the performance of different processing techniques with respect to raw (unprocessed data) and the performance of uncooled vs. cooled cameras.

A UAV equipped with a thermal camera was used for an IR active thermography test on a composite wing. The UAV was tracked, and the movement was within a few centimetres. The demonstration showed that the UAV needs to maintain a specific 3D location whilst capturing IR data. This allows for adequate post-processing. The flight was manually flown and, even though it did locate the subsurface debris, the data were noisy, which resulted in low thermal contrast, making it difficult to process the signal. Future work will aim to minimise the movement of the in-flight UAV whilst capturing a thermal sequence, which will be done using the onboard computer to send localisation commands to the UAV with the help of the Vicon system, as explained in Section 5. A stable flight will result in a less noisy dataset. There are other sensors such as LiDAR, stereovision, and ultrasonic sensors, which can help with UAV localisation.

## Figures and Tables

**Figure 1 sensors-20-03381-f001:**
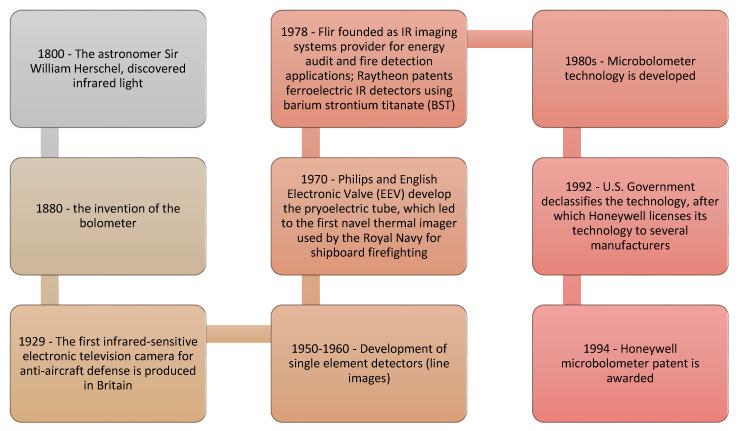
Timeline of infrared (IR) technology [7].

**Figure 2 sensors-20-03381-f002:**
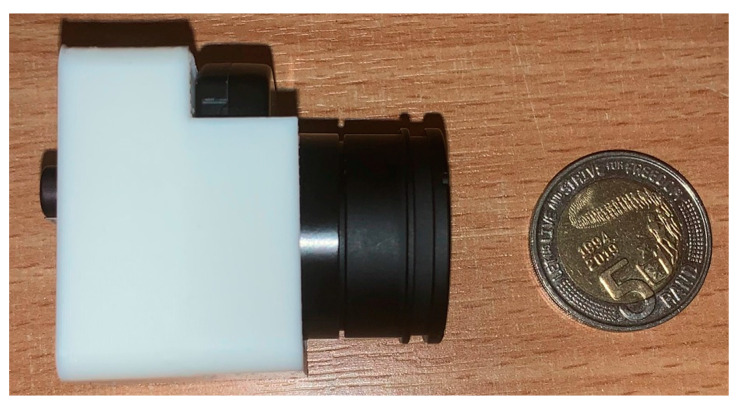
FLIR Boson In a 3D printed case next to a coin for scale, the camera weighs around 7.5 g.

**Figure 3 sensors-20-03381-f003:**
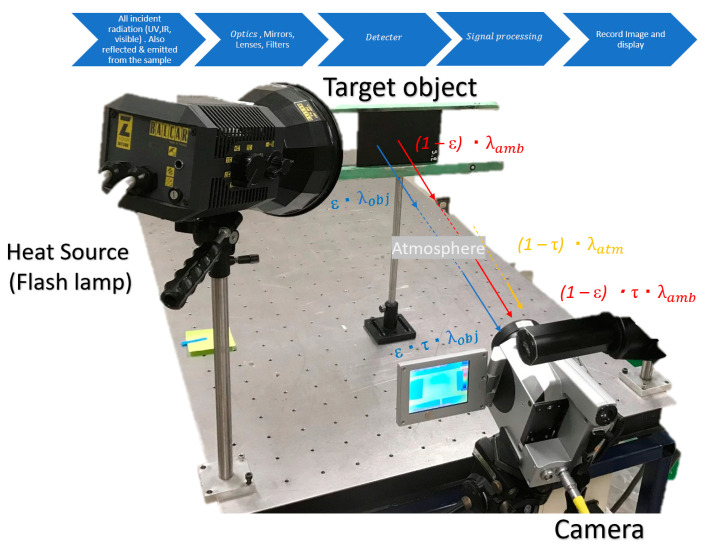
Visualisation of the inspection with regards to the emitted and absorbed radiation. The blue arrows display the process of the IR camera [11].

**Figure 4 sensors-20-03381-f004:**
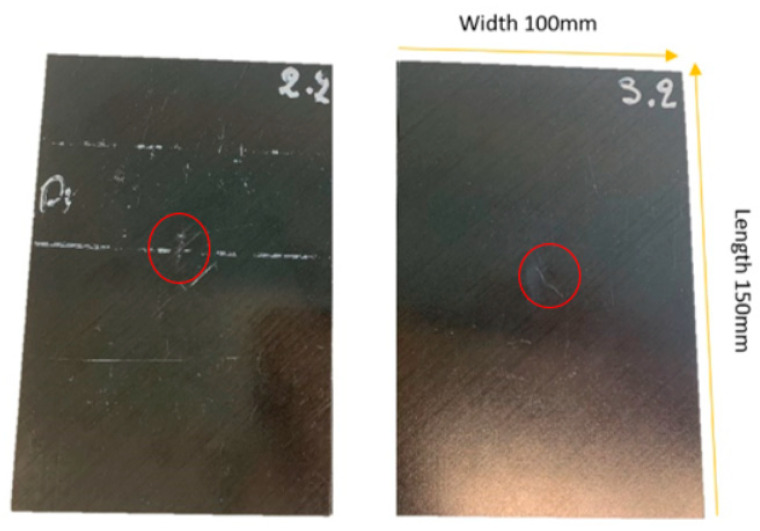
Images of samples 2.2 and 3.2. The samples were manufactured according to the American Society for Testing and Materials (ASTM) standard D7136. The layup is quasi-isotropic (45/0/45/90) with a thickness of 4.2 mm. The red circle highlights the location of the impact from the drop tower. Reflection of the camera taking the image can also be seen.

**Figure 5 sensors-20-03381-f005:**
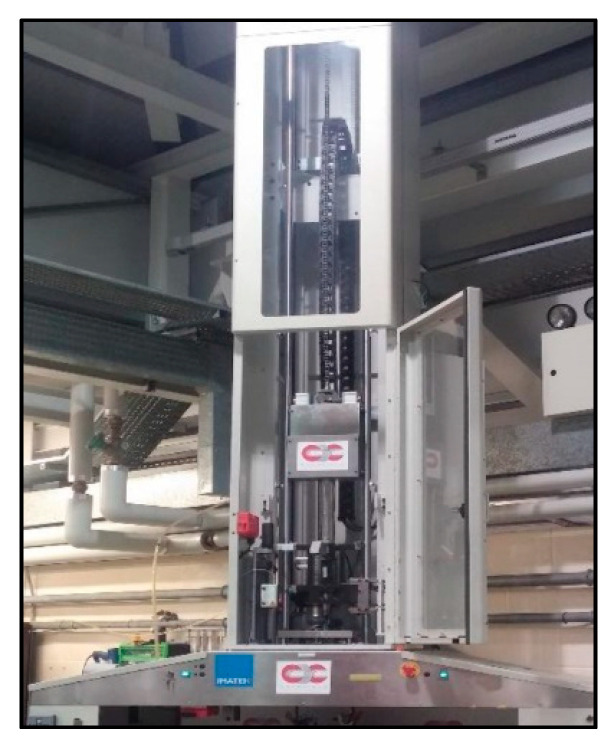
The drop weight tower.

**Figure 6 sensors-20-03381-f006:**
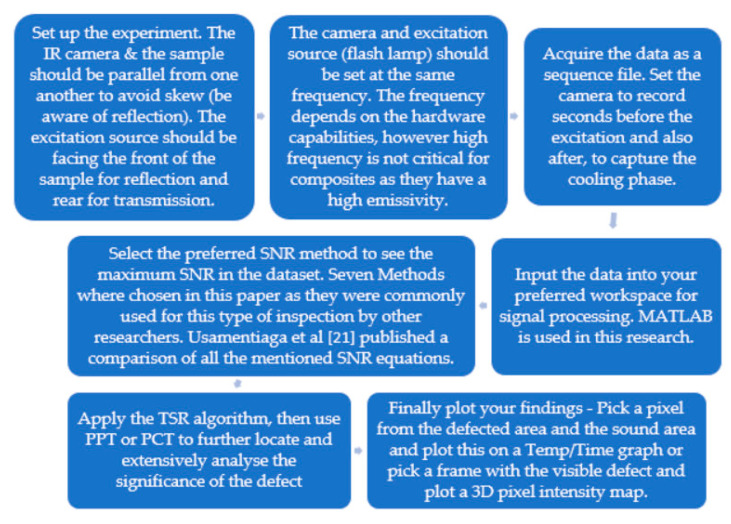
The method strategy according to an active thermographic test.

**Figure 7 sensors-20-03381-f007:**
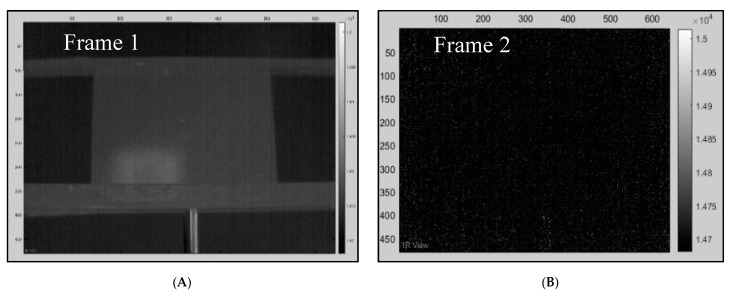
Dataset from sample 3.2, front surface, using the uncooled camera. (**A**) Reflection; (**B**) Post cold image subtraction, displaying just noise.

**Figure 8 sensors-20-03381-f008:**
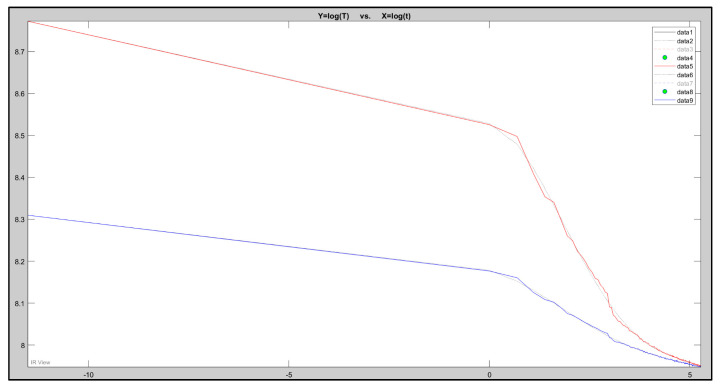
Five-degree polynomial fitting for sample 3.2 front surface (red—defected area pixel; blue—sound area pixel).

**Figure 9 sensors-20-03381-f009:**
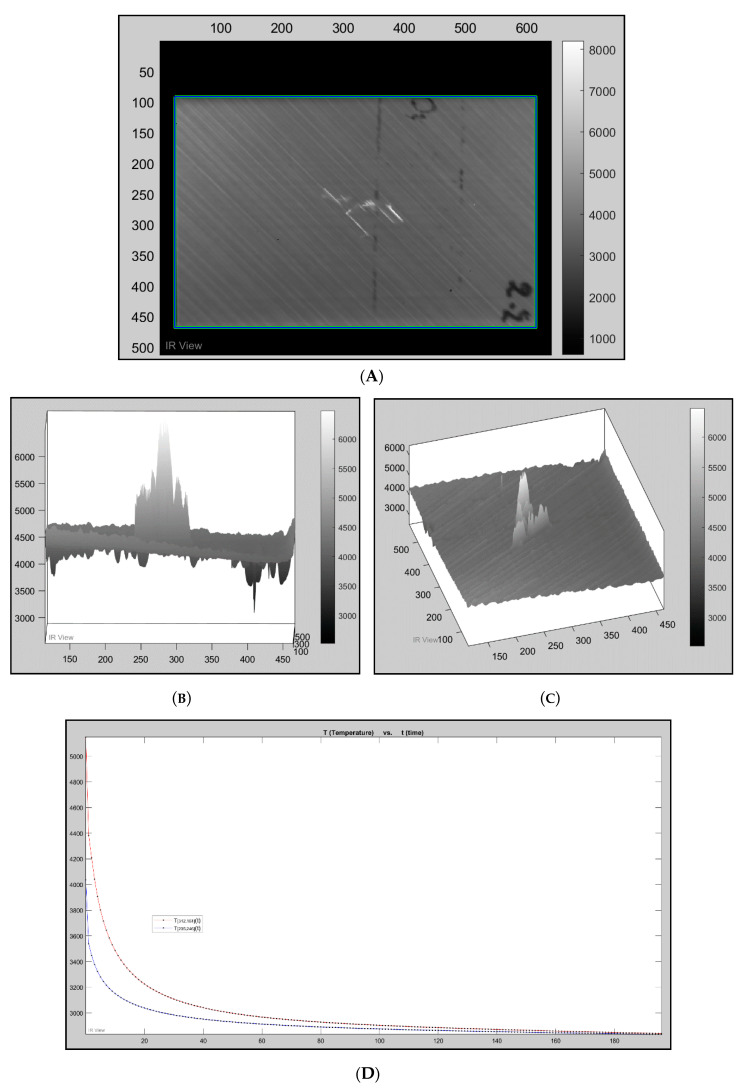
(**A**) *2.2CooledFront*, frame 4, where the flash hit the specimen. (**B**) *2.2CooledFront*, frame 4, where the flash hit the specimen. Three-dimensional (3D) pixel map side angle. (**C**) *2.2CooledFront*, frame 4, where the flash hit the specimen. 3D pixel map top angle. (**D**) Temperature vs. time graph; the black dots each represent a frame (blue—sound pixel; red—defected pixel).

**Figure 10 sensors-20-03381-f010:**
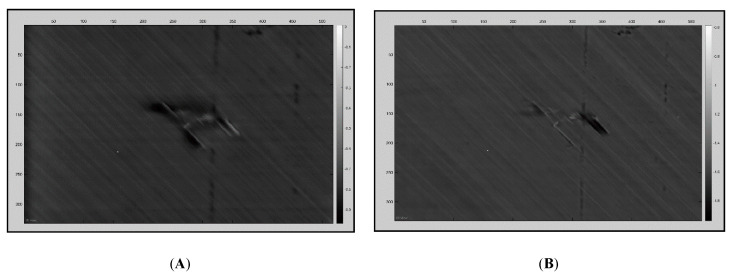
(**A**) Fourier transform phase; the damage shown is the subsurface. (**B**) Fourier transform phase; the damage visible is closer to the surface.

**Figure 11 sensors-20-03381-f011:**
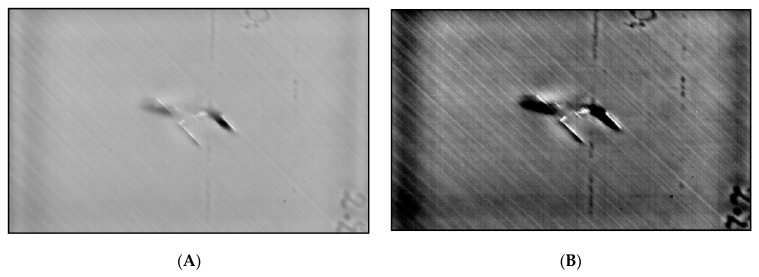
(**A**) Post-processed principal component technique (PCT) thermal output data. (**B**) The contrast difference can be made clearer using colour thresholding to help visualise the real significance of the impact damage.

**Figure 12 sensors-20-03381-f012:**
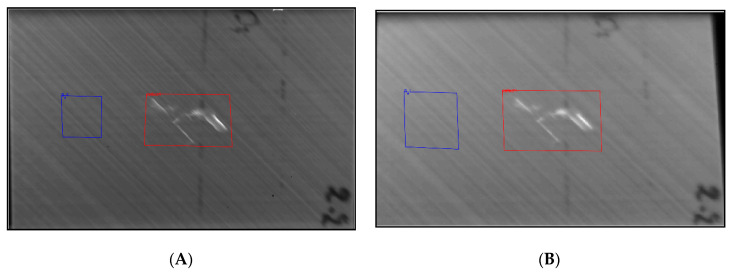
(**A**) *2.2.Cooled Front*, selecting areas of the damaged and sound areas. (**B**) *2.2.Uncooled Front*, selecting areas of the damaged and sound areas.

**Figure 13 sensors-20-03381-f013:**
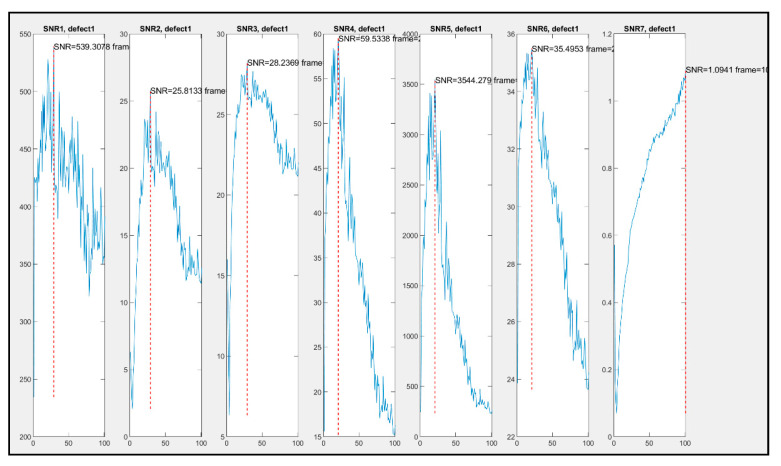
Seven signal-to-noise ratio (SNR) calculations from data *2.2.*Cooled Front. All other SNR graphs are available but not included for simplification.

**Figure 14 sensors-20-03381-f014:**
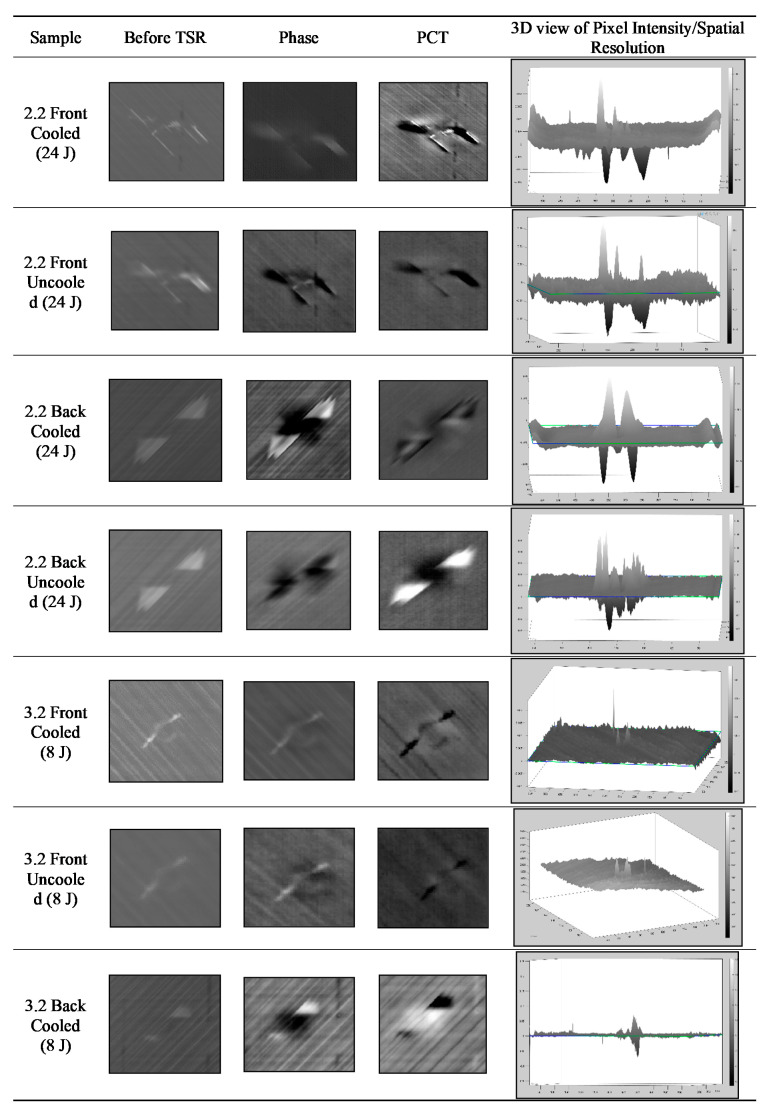
(**A**) All Active thermographic data from raw to post-processed data. (**B**) Enlarged images outlining the defects and displaying the size in both samples.

**Figure 15 sensors-20-03381-f015:**
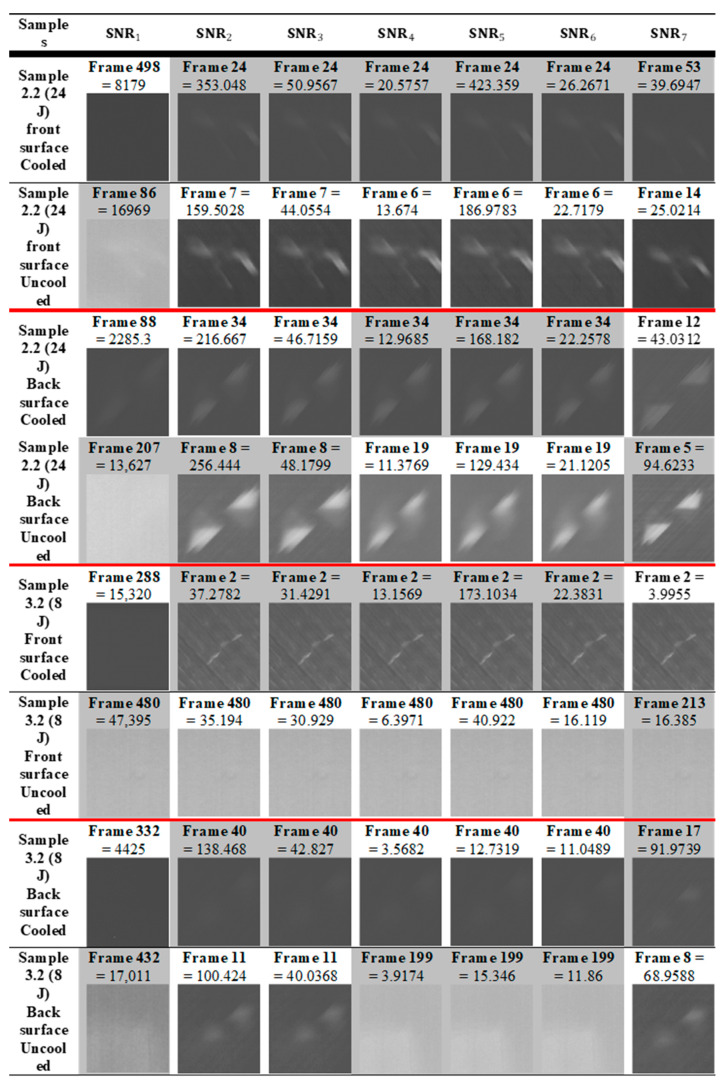
SNR equations 1–7 of the raw dataset from both the cooled and the uncooled IR camera. The numerical values are output after the SNR calculations from the pixel intensity regions selected within the defect and sound areas. The shaded boxes display the higher SNR value from the sample of the cooled and uncooled dataset.

**Figure 16 sensors-20-03381-f016:**
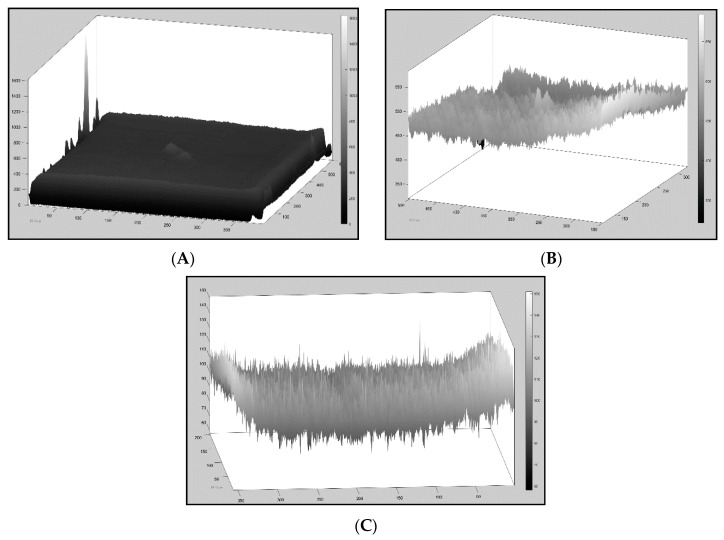
(**A**) 3D pixel map of sample 3.2 back surface cooled, frame 40. (**B**) 3D pixel map of sample 3.2 back surface uncooled, frame 11. (**C**) 3D pixel map of sample 3.2 back surface uncooled, frame 199.

**Figure 17 sensors-20-03381-f017:**
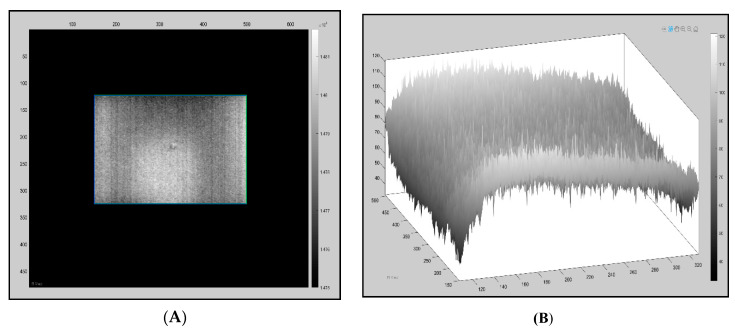
(**A**) Focused image of frame 480 of sample 3.2uncooledfront surface. The damage is visible in the noisy dataset which also contains some reflectance. (**B**) 3D pixel map of sample 3.2uncooledfront surface, frame 480. (**C**) Temperature/time graph from 3.2 front surface uncooled (red—defect area pixel; blue—sound area pixel).

**Figure 18 sensors-20-03381-f018:**
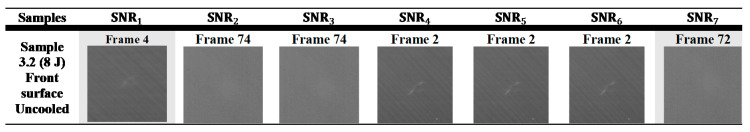
The SNRs for 3.2frontuncooled originally failed to locate the defect; however, after TSR, ${\mathrm{SNR}}_{4,5,6}$ successfully located the defect.

**Figure 19 sensors-20-03381-f019:**
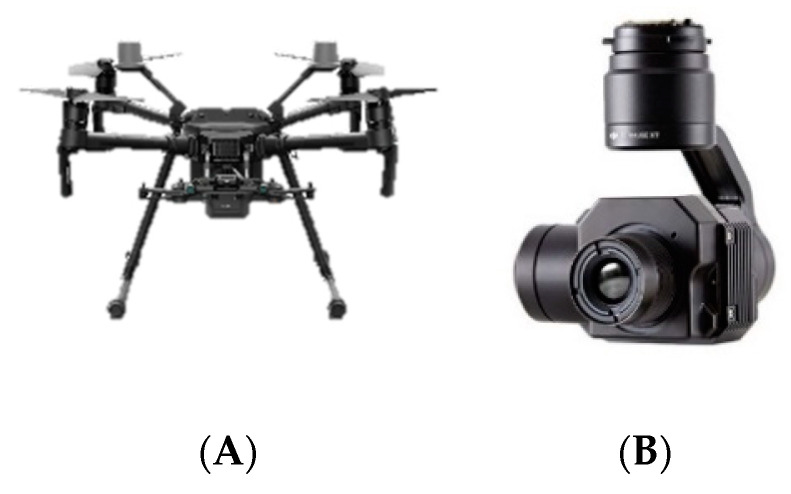
(**A**) DJI M210. (**B**) Zenmuse XT.

**Figure 20 sensors-20-03381-f020:**
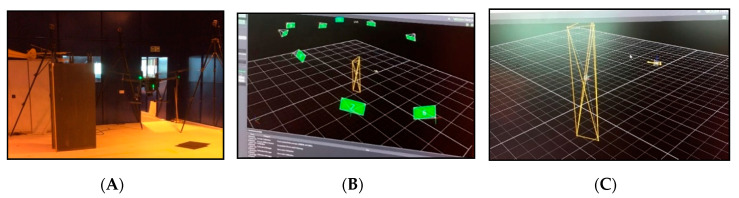
(**A**) DJI M210 inspecting the composite wing. (**B**) Vicon tracking system. (**C**) The wing and the unmanned aerial vehicle (UAV) being tracked.

**Figure 21 sensors-20-03381-f021:**
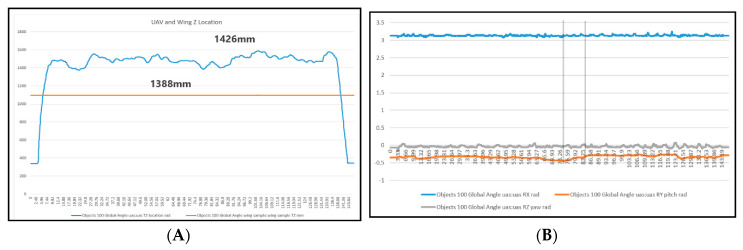
(**A**) The *z*-axis shows the flight envelope: 1426 mm − 1388 mm = 38 mm. (**B**) Graph demonstrating the stability of the UAV in flight. This was measured by the roll along the *x*-axis, pitch along the *y*-axis, and yaw along the *z*-axis.

**Figure 22 sensors-20-03381-f022:**
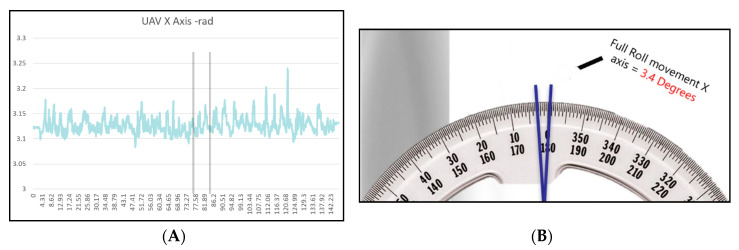
(**A**) Graph displaying the UAV roll along the *x*-axis. (**B**) Image displaying the full roll of the UAV. It rolled within a 3.4° envelope along the *x*-axis.

**Figure 23 sensors-20-03381-f023:**
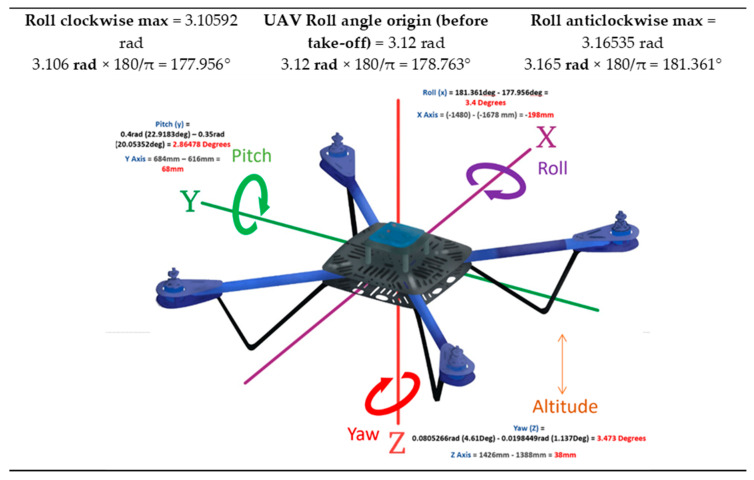
Manual full flight envelope whilst capturing 6 s worth of data. The movement is significant and has a direct effect on the NDT data captured using a thermal camera [26].

**Figure 24 sensors-20-03381-f024:**
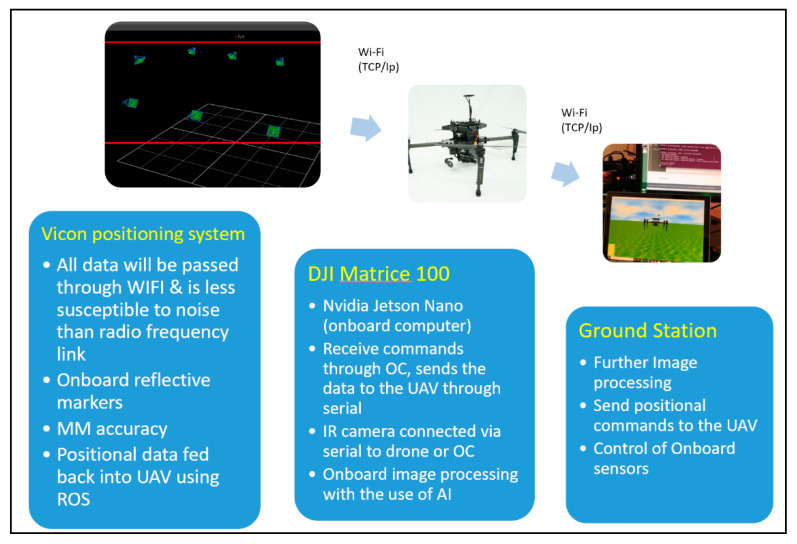
Flow chart of the indoor inspection process.

**Figure 25 sensors-20-03381-f025:**
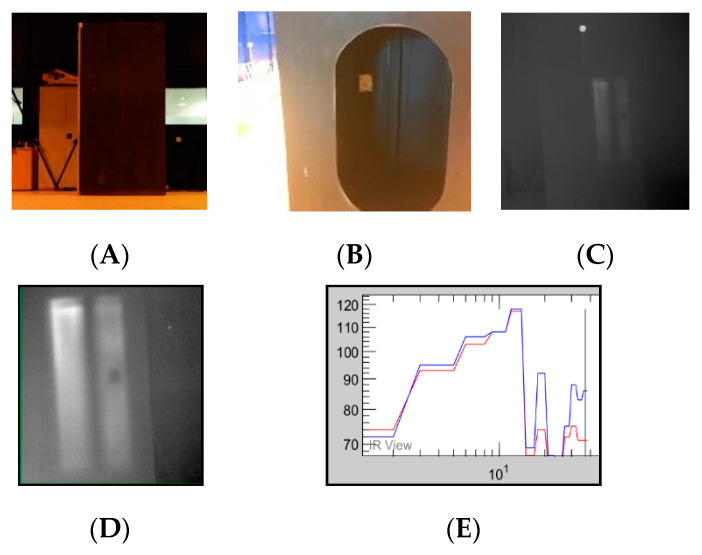
(**A**) Red/green/blue (RGB) image of the composite wing. (**B**) Rear image of the wing displaying the debris inside. (**C**)IR image taken of the composite wing using the Zenmuse XT camera. (**D**) Debris prompted a contrast difference in the thermal data. (**E**) Temperature/time graph. The images where taken at different locations. Interference and instability resulted in corrupted data.

**Figure 26 sensors-20-03381-f026:**
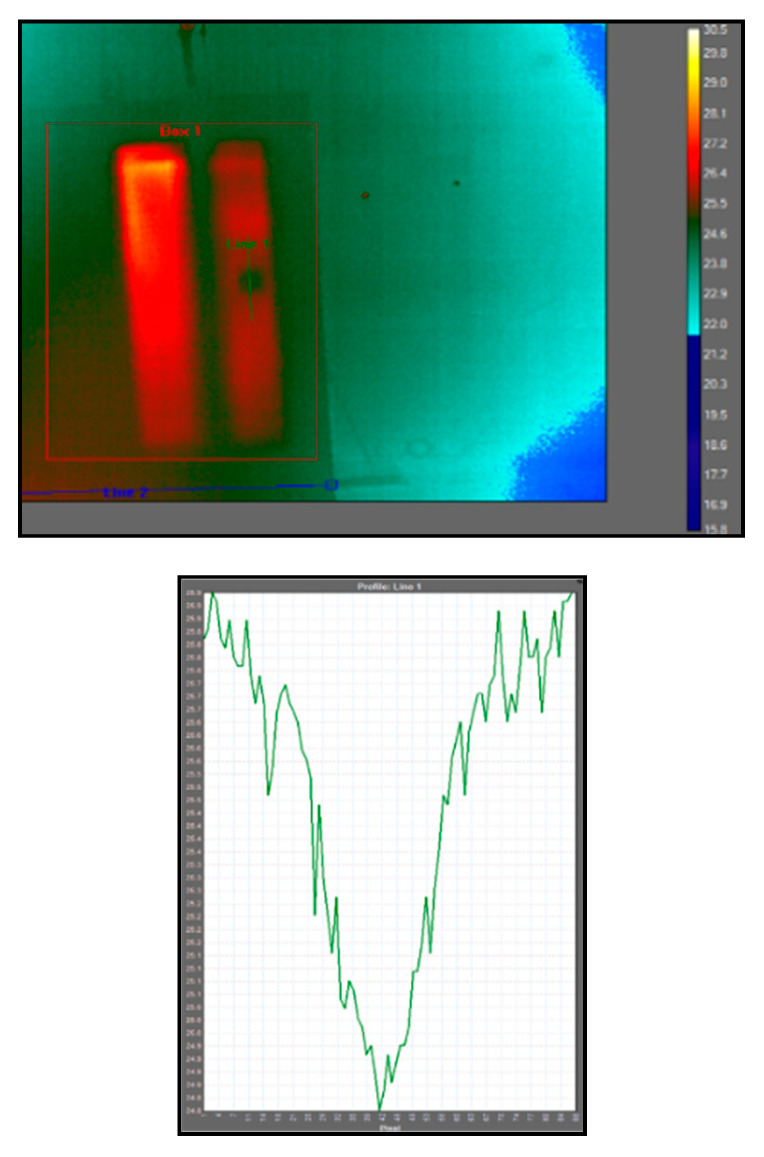
Using research IR with FLIR, the data could be further processed. The profile line displays the pixel contrast difference where the “defect” is.

**Figure 27 sensors-20-03381-f027:**
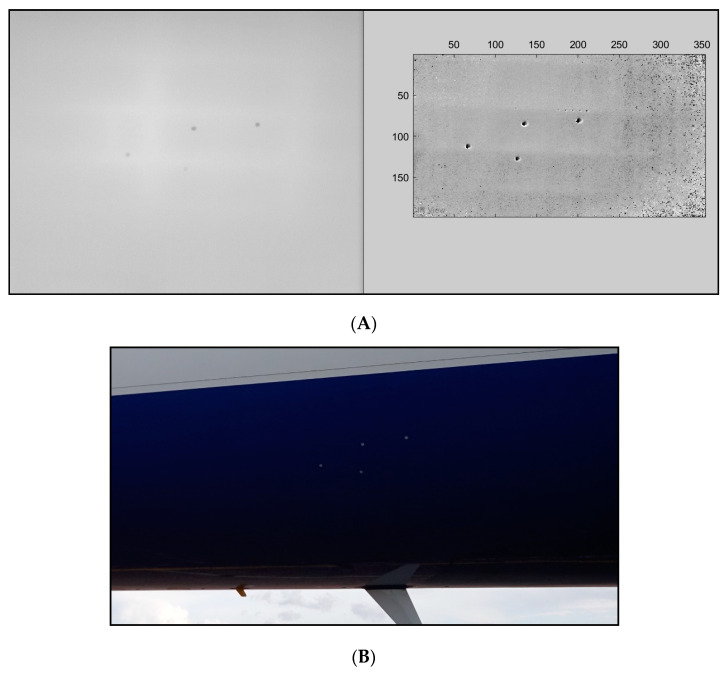
(**A**) Raw IR image of the B737 fuselage after signal processing. (**B**) RGB image of the B737 fuselage.

**Figure 28 sensors-20-03381-f028:**
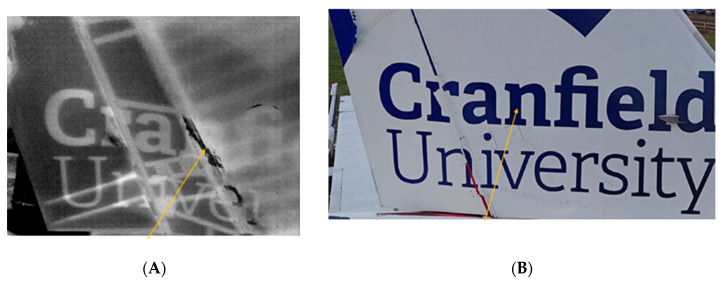
(**A**) Passive thermographic image of the tail/rudder of a Boeing 737. (**B**) Corresponding RGB image of the Boeing 737.

**Table 1 sensors-20-03381-t001:** The table compares the specs of the two cameras which are used in the active thermographic non-destructive test of the aerospace grade composites [14,15]. FPA—focal plane array; A/D—analogue to digital; IEC—International Electrotechnical Commission; RMS—root mean square; N/A—not applicable.

	Jenoptik VarioCAM High Resolution	FLIR Phoenix
Spectral range	LWIR (7.5–14) μm	MWIR (3–5) μm
Pixels	640 × 480	640 × 512
Detector	Uncooled microbolometor FPA	InSb (indium antimonide) FPA
Start-up time	<60 s	10–15 min
IR frame rate (full frame)	50/60 Hz	Up to 90 Hz
A/D conversion	16 bit	14 bit
Power supply	Battery (3 h)	Cabled power supply
Vibration resistance in operation	2 G, IEC 68-2-6	6.7 g, RMS random vibe, all 3 axis
Size (L × W × H)	(133 × 106 × 110) mm	(190.5 × 111.8 × 132.1) mm
Weight	1.5 kg (completely equipped)	3.2 kg (excluding lens)
Cooling engine	N/A	Stirling closed cycle (~77 K)
Noise-equivalent temperature difference (NETD) performance	70 mK	25 mK
Integration time (electronic shutter speed)	N/A	9 µs to full frame time
Lens	25-mm lens Field of view = 30° × 23°	50-mm lens Field of view = 18° × 15°

**Table 2 sensors-20-03381-t002:** Common examples of impact damage cases on a commercial aircraft [5].

Impact Damage	Energy	Mass	Velocity	Hard/Soft Body Impact
Hail in flight	37 J	0.001 kg	37 m/s	Soft
Bird strike	59 kJ	3.63 kg	180 m/s	Soft
Engine debris	182 kJ	2.72 kg	366 m/s	Hard
Rim fragment	8.4 kJ	1.68 kg	100 m/s	Hard
Tire fragment	5.2 kJ	2.45 kg	64 m/s	Hard
Runway debris	>20 J	0.01 kg	>60 m/s	Hard
Tool drop	28 J	0.56 kg	10 m/s	Hard
Hail on ground	100 J	0.113 kg	42 m/s	Hard

**Table 3 sensors-20-03381-t003:** M21 resin benefits from toughening low-volume percentage rubber particles with a mean diameter of 20–40 μm. Hexcel^®^ M21 curing data sheet. Sourced from Reference [16]. ILSS—interlaminar shear strength.

Ply Mass	Ply Thickness	Fibre Volume Fraction	ILSS
305 g/m^2^	0.262 mm	56.6%	60 MPa
Tensile strength	Tensile modulus	*G* _Ic_	*G* _IIc_
3039 MPa	172 GPa	765 J/m^2^	1250 J/m^2^

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
