# Peer review of "Comparison of Cooled and Uncooled IR Sensors by Means of Signal-to-Noise Ratio for NDT Diagnostics of Aerospace Grade Composites"

_sensors, 2020, doi:10.3390/s20123381_

Round 1

Reviewer 1 Report

This paper present cooled and uncooled IR sensors by  means of signal-to-noise ratio for NDT diagnostics of  aerospace grade composites in detail. The concept of using a UAV for NDT on a composite wing is explored and the UAV is also tracked using a localisation system to observe the exact movement in millimetres and how it affects the thermal data. Here are some suggestions:

1. This paper introduces 7 SNR methods, and asks the author to give the method selection strategy according to the test conditions.

2. The UAV and the composite sample were tracked using a VICON localisation system. How to make inquiry the influence of uav's stability on test system and test results.

3. It is suggested to add an impact test sample between 8J and 24J for analysis.

Author Response

Please see the attachments for responses, including tracked changes.

thanks

Reviewer 2 Report

Although the idea of this study is interesting, the execution is relatively poor and must be improved in many ways.

  1. I suggest using symbol small lambda for wavelength.
  2. Figure 1: Sir William Herschel, manufacturers.
  3. Line 135: two words "under". Consider revising.
  4. Line 193: What is meant by temperature coefficient? What kind of quantity is that?
  5. Figure 6: check grammar of label.
  6. Line 237: the reference [26] in not in the reference list.
  7. Table 1 is not mentioned in the main text.
  8. Subsection 2.3: the concept of cooled cameras is not described, although it is used in subsequent portions of text.
  9. Figure 8A is not necessary, it does not serve any practical purpose.
  10. Pictures of both composite panels clearly indicating position of damage are missing.
  11. Why is there no scale attached to axes in Figure 9B?
  12. Figure 9B: authors could have made a 3D surface plot showing temperature vs time vs coordinate of the whole region of damage, not just 1 pixel.
  13. Subsection 4.1.1 is called "Defect analysis", but, as it is, there is not much analysis in there. Can the temperature decays be fitted with a functional relationship? That way, at least the decay rates of temperature could be compared between healthy an damaged regions. The same holds true for Figure 15A.
  14. I suggest to reverse the coloring and use red for damaged and blue for sound.
  15. Why are Figures 10 and 11 positioned together in one row?
  16. Figures 10-12 are not mentioned in the text, nor are they described.
  17. Equation 1: use subscripts, not superscripts. Superscripts may be confused with powers.
  18. Is the noise defined as RMS of pixel values (in healthy and damaged regions)?
  19. Figures 13 not mentioned in the text.
  20. Figure 14: What quantity is shown on the abscissa axis?
  21. Figure 15A not mentioned in the text. what is the difference between 2 columns of phase and 2 columns of PCT? Graphs of temperature decay are too small to see all the labels.
  22. Figure 15B: there are two 3.2 Front (8J) images.
  23. table 3 and Figure 17: authors should provide some numerical value for each SNR so that the different representations of SNR can be compared objectively.
  24. Defect localization should using drones should be performed outdoors in less controlled environments to see the real performance of the approach.

Overall, the manuscript can not be accepted. It feels very raw and rushed. The analysis part is weak, many explanations are missing. There is no validation in real environments.

Author Response

Please see attachment including tracked changes.

thanks

Round 2

Reviewer 1 Report

The author has revised the paper according to the comments of the review, and it is suggested to be accepted.

Reviewer 2 Report

The manuscript has been improved by adding more description and clarification with respect to the original submission. However, some issues still remain.

1) It is still not clear what mathematical model was applied to describe the temperature decay in the composite specimens.

2) Is it possible to foresee which definition of SNR will perform better in each case? For example, in metal, ceramic or composite structures which have different surface textures and overall material properties? Or is it necessary to test all variants of SNR and select the best for each case separately? In other words, is there a "one size fits all" SNR recipe or not?

3) What are the possibilities of using the drones equipped with IR camera for detecting damage outdoors? Will the temperature differences not mask the presence of damage? This practical problem and other problems have to be addressed in the discussion part.

4) Literature review seems a little lacking. Consider referring to more recent studies

In light of all this, the paper still has to be revised and cannot be accepted for publication in it's current state.
